# Novel Nuclear Medicine Imaging Applications in Immuno-Oncology

**DOI:** 10.3390/cancers12051303

**Published:** 2020-05-21

**Authors:** Stefano Frega, Alessandro Dal Maso, Giulia Pasello, Lea Cuppari, Laura Bonanno, PierFranco Conte, Laura Evangelista

**Affiliations:** 1Oncology 2 Unit, Department of Medical Oncology, Veneto Institute of Oncology IOV – IRCCS, 35128 Padova, Italy; giulia.pasello@iov.veneto.it (G.P.); laura.bonanno@iov.veneto.it (L.B.); pierfranco.conte@unipd.it (P.C.); 2Department of Surgery, Oncology and Gastroenterology, University of Padua, 35128 Padova, Italy; alessandro.dalmaso@iov.veneto.it; 3Nuclear Medicine Unit, Veneto Institute of Oncology IOV–IRCCS, University of Padua, 35128 Padova, Italy; lea.cuppari@gmail.com; 4Nuclear Medicine Unit, Department of Medicine, University of Padua, 35128 Padova, Italy; laura.evangelista@aopd.veneto.it

**Keywords:** nuclear medicine, positron-emission tomography, single-photon emission computed tomography, immunotherapy, immune checkpoint inhibitors

## Abstract

The global immuno-oncology pipeline has grown progressively in recent years, leading cancer immunotherapy to become one of the main issues of the healthcare industry. Despite their success in the treatment of several malignancies, immune checkpoint inhibitors (ICIs) perform poorly in others. Again, ICIs action depends on such a multitude of clinico-pathological features, that the attempt to predict responders/long-responders with ad-hoc built immunograms revealed to be quite complex. In this landscape, the role of nuclear medicine might be crucial, with first interesting evidences coming from small case series and pre-clinical studies. Positron-emission tomography (PET) techniques provide functional information having a predictive and/or prognostic value in patients treated with ICIs or adoptive T-cell therapy. Recently, a characterization of the tumor immune microenvironment (TiME) pattern itself has been shown to be feasible through the use of different radioactive tracers or image algorithms, thus adding knowledge about tumor heterogeneity. Finally, nuclear medicine exams permit an early detection of immune-related adverse events (irAEs), with on-going clinical trials investigating their correlation with patients’ outcome. This review depicts the recent advances in molecular imaging both in terms of non-invasive diagnosis of TiME properties and benefit prediction from immunotherapeutic agents.

## 1. Immuno-Oncology (I-O)

The host immune system interacts with tumor cells through the activation of innate and adaptive immune mechanisms. Initially, transformed cells are eliminated by a competent immune system. Nevertheless, sporadic tumor cells manage to survive immune destruction and may enter an equilibrium phase during which editing occurs. Finally, immunologically sculpted tumors begin to grow progressively, establish an immunosuppressive tumor microenvironment and become clinically apparent [1]. The immune system–tumor interaction is targeted by immunotherapy, with the aim of stimulating the immune system to direct immune-mediated responses against the tumor.

The global I-O pipeline has grown progressively in last years. The administration of interleukin-2 (IL-2) and the adoptive transfer of antitumor T cells grown in IL-2 represented the first effective immunotherapies for cancer in humans [2]. Toll-like receptor agonist imiquimod has been used to treat a variety of skin cancers including basal cell cancer, squamous cell cancer, lentigo maligna melanoma, and cutaneous T-cell lymphoma [3]. Recombinant interferon α-2b is indicated in hairy cell leukemia, chronic myelogenous leukemia, multiple myeloma, follicular lymphoma, and as adjuvant therapy in malignant melanoma [4]. Calmette-Guerin bacillus is the gold standard adjuvant treatment of high-risk non-muscle invasive bladder cancer [5].

Moreover, the development of immune checkpoint inhibitors (ICIs) is a revolutionary milestone in the field of I-O. ICIs reinvigorate antitumor immune responses by interrupting co-inhibitory signaling pathways and promote immune-mediated elimination of tumor cells. Anti-programmed-death 1 (PD-1) nivolumab and pembrolizumab, anti-programmed death-ligand 1 (PD-L1) atezolizumab, durvalumab and avelumab, and anti-cytotoxic T-lymphocyte antigen 4 (CTLA-4) ipilimumab are the standard of care in adjuvant treatment or advanced disease treatment for many solid tumors (Table 1).

However, despite these substantial advances, only a subset of patients receiving ICIs derive clinical benefit. It is then crucial to identify and to develop predictive biomarkers of ICIs response. PD-L1 expression and tumor mutation burden are the only predictive factors validated in phase III clinical trials, but are still imperfect since there are ICI non-responders expressing high biomarkers levels and ICI responders with low biomarkers levels. New determinants of response are being investigated, with strategies encompassing multiple biomarkers [49,50].

## 2. Cancer, Nuclear Medicine (NM) and Response to I-O

### 2.1. Nuclear Medicine

Nuclear medicine is a branch of medicine using radionuclides in the diagnosis and treatment of diseases. Diagnostic applications consist in “functional imaging” and are based on the ability of radiopharmaceutical agents to concentrate in pathological tissues and to emit radiations that are revealed by external detectors and then recorded. Therapeutic approaches exploit selective concentration of radiopharmaceutical agents in pathological tissues and the use of the radiations to destroy them. Techniques of nuclear medicine imaging include two-dimensional scintigraphy, three-dimensional single-photon emission computed tomography (SPECT) and positron emission tomography (PET), and hybrid techniques combining computed tomography (CT) or magnetic resonance imaging (MRI) scans with SPECT or PET [51].

2-[^18^F]fluoro-2-deoxy-d-glucose (^18^F-FDG or FDG) PET/CT has become an established standard nuclear imaging modality in oncology [52] based on the finding that tumor hypoxic cells have increased glucose demand and, consequently, FDG uptake and accumulation [53]. FDG-PET/CT is thus widely used in cancer diagnosis, staging, and response to treatment evaluation [54,55,56].

However, some drawbacks have to be mentioned [57]. Infection and inflammation can lead to false positive findings. Another limitation is that some organs display a higher physiological FDG uptake (i.e., brain, myocardial tissue, brown fat) and therefore are not studied with this modality. Moreover, FDG-PET/CT can result false negative for lesions or <8–10 mm and for several histotypes. Hyperglycemia has to be correctly managed to convey a valid exam. Furthermore, FDG-PET/CT is nonspecific for tumor subtypes. Finally, risk–benefit considerations should minimize radiation exposure. For all these reasons, clinical information exchange between oncologists and nuclear medicine specialists is crucial for patients’ care and outcomes [54,55,56].

### 2.2. Assessment of Tumor Response

Assessment of response represents a measure of antitumor drug activity. Radiological overall response rate is the primary endpoint of about 70% of phase II trials. Thus, positive trials are the basis of accelerated approval in refractory advanced disease, a setting where there is no available therapy [58]. Furthermore, response assessment is a cornerstone of daily clinical practice.

Practically, response assessment starts with summing the size of tumor lesions in a baseline CT scan before beginning a new therapy. After a specified time, the relative change of the sum of lesion sizes is registered. Finally, a threshold on this relative change is applied to decide whether treatment should be administered or discontinued. Historically, the WHO Response Criteria were first published in 1981 and were based on bidimensional measurements [59]. As CT scans were more easily available, the WHO Criteria were simplified and harmonized by the Response Evaluation Criteria in Solid Tumors (RECIST) in which unidimensional measurements were considered [60]. A further update was provided by the RECIST 1.1 version, including revised specifications on measurable disease, nodal disease, and progressive disease [61].

Nevertheless, less than half of the studies that met their response endpoint measured with RECIST criteria in a phase II clinical trial are successful in phase III clinical trials [62,63,64]. It has been proposed that response assessment with FDG-PET could be more sensitive and specific for the detection of metastatic disease. In particular, bone disease is not evaluable by RECIST criteria. Moreover, stable disease as defined by RECIST criteria could represent on the one hand a consequence of tumor growth inhibition by effective therapy, on the other hand a result of an indolent tumor biology, whereas it is more likely that growth inhibition results in a decrease of tumor FDG uptake [65].

Two metabolic response criteria were thus developed. European Organization for Research and Treatment of Cancer (EORTC) PET response criteria were based on standalone PET imaging and focused on evaluation of only one lesion [66]. PET Response Criteria in Solid Tumors (PERCIST) were applied to PET/CT scans and deal with multiple lesions evaluation [67,68]. Moreover, main differences include the use of standardized uptake value (SUV) mean in EORTC criteria compared to the SUV peak normalized by lean body mass (SULpeak) used in PERCIST criteria as the quantitative parameter, finally a different threshold for progressive metabolic disease is used. Robust clinical validation is still awaited.

### 2.3. Assessment of Tumor Response to Immunotherapy

Response assessment for I-O must deal with a unique challenge, pseudoprogression, that is a temporary increase in tumor size, meeting the criteria for progressive disease based on RECIST, which is lately not confirmed. Pseudoprogression is caused by immune cells infiltrating tumor lesions. Its prevalence is up to 10%, and it has been characterized in malignant melanoma studies [69,70].

Four new response criteria have been proposed to solve this issue and all of these require a confirmation of progressive disease [71]. Immune-related response criteria (irRC) are based on WHO criteria, and therefore on bidimensional measurements [72]. Immune-related response evaluation criteria in solid tumors (irRECIST) combine irRC with RECIST 1.1 criteria, thus rely on unidimensional measurements [73]. Immune response evaluation criteria in solid tumors (iRECIST) are based on RECIST 1.1, too, but differ from irRECIST because they do not include measurements of the new lesions on tumor burden [74]. Immune-modified response evaluation criteria in solid tumors (imRECIST) are similar to irRECIST [75].

IrRC and irRECIST were clinically validated in patients with advanced malignant melanoma treated with ipilimumab, imRECIST in patients with advanced NSCLC and metastatic urothelial carcinoma treated with atezolizumab, whereas iRECIST are based on a consensus statement. The RECIST working group recommends that recent trials should use RECIST 1.1 to define the primary and secondary efficacy-based endpoints and reserve irRC or irRECIST for exploratory endpoints [74].

The role of already published metabolic criteria was investigated in patients treated with immunotherapy. EORTC criteria applied to PET/CT evaluation after two cycles of ipilimumab on 22 advanced melanoma patients were found to be predictive of final treatment outcome [76]. Retrospective baseline and 1-year PET/CT analysis with EORTC criteria of 104 metastatic melanoma patients treated with nivolumab or nivolumab and ipilimumab showed that patients with complete metabolic responses at 1 year had ongoing responses to therapy thereafter [77]. PERCIST criteria were used to evaluate response on a PET/CT scan 1 month after the start of nivolumab in 24 advanced NSCLC patients, metabolic responses were closely associated with therapeutic response and survival [78].

Likewise, semi quantitative parameters were found to be predictive of outcome in patients treated with immunotherapy. This could be the case for SUV, metabolic tumor volume (MTV), and tumor total lesion glycolysis (TLG). These three variables define the entire metabolic tumor burden (MTB), that may represent a reliable value to predict treatment success, being known that I-O work better in a context of low tumor load [79]. Variations in MTV and TLG in FDG-PET/CT scans taken before and after 4 cycles of nivolumab in 20 patients with metastatic NSCLC correlated with objective response by CT scan [80]. Baseline SUVmax and SUVmean identified progression after 8 weeks of therapy in 27 NSCLC patients treated with nivolumab or pembrolizumab [81]. Increased SUVmax and SULpeak of the most FDG-avid lesion at interim PET/CT identified non-responders in 30 advanced melanoma patients treated with pembrolizumab [82]. Whole-body maximum SUV (SUVmaxwb) in 32 advanced NSCLC patients treated with nivolumab correlated with response [83]. Moreover, an associated flare response on FDG PET/CT imaging at 1 week post-therapy defined as >100% increase in tumor SUVmax between baseline and post-therapy scans was found to be predictive of a complete response to immunotherapy, and this correlated usually with evidence of T cell reinvigoration in the blood at 1 week post therapy [84].

In parallel, several I-O-adapted metabolic criteria have been proposed. PET/CT Criteria for Early Prediction of Response to Immune Checkpoint Inhibitor Therapy (PECRIT) are based on RECIST 1.1 to define complete response, partial response and progressive disease, whereas percent change in SULpeak by PERCIST criteria is used to distinguish patients with stable disease per RECIST 1.1 and clinical benefit from those with stable disease and no clinical benefit. The comparison of baseline PET/CT scan with a second PET/CT scan at 3–4 weeks after the start of immunotherapy based on PECRIT was able to predict the best overall response at 4 months in 20 patients with advanced melanoma patients treated with nivolumab, ipilimumab or BMS-936559 [85].

Conversely, the number of new FDG-avid lesions was considered in PET Response Evaluation Criteria for Immunotherapy (PERCIMT). Four newly emerged avid lesions on PET/CT scan after 3 months of therapy was found to be predictive of treatment failure in 41 melanoma patients treated with ipilimumab, with the functional size of the new lesions playing an important role in predicting the clinical response [86]. PERCIMT evaluation was applied to PET/CT scan after two cycles of immunotherapy on the same cohort of patients, showing higher sensitivity compared to EORTC criteria in predicting clinical benefit, but specificity in predicting no clinical benefit was not significantly higher [87].

Inflammatory response caused by immunotherapy may confound the ability of FDG-PET to identify patients with a favorable response. Thus, similarly to iRECIST, immune PET Response Criteria in Solid Tumors (iPERCIST) included the concept of disease progression confirmation in PERCIST. iPERCIST were retrospectively applied to a cohort of 28 NSCLC patients treated with nivolumab. All patients underwent a baseline PET/CT scan and a second scan after 2 months of treatment, which was evaluated with the proposed criteria. A metabolic unconfirmed progressive disease was re-evaluated with a third PET/CT scan after 4 weeks. The 1-year survival rates were significantly higher in iPERCIST responders compared with non-responders [88]. Beyond the above seen parameters, even spleen to liver ratio (SLR) has been found to correlate with ICIs efficacy, and thus may represent a novel PET biomarker to evaluate patients’ baseline immune state prior to immunotherapy [89]. All these criteria require further validation.

Figure 1 depicts an overview of CT-based and PET-based response criteria for solid tumors.

## 3. NM and Tumor Immune-Microenvironment (TiME) Analysis

The survival benefit from using ICIs varies considerably from patient to patient. ICIs manage to slow down cancer progression, changing completely long-term survival perspective for a subgroup of patients, that goes from 10% to 30%, depending from the tumor types and drug used [90,91]. The reverse of the medal, of course, is that ICIs own poor or no efficacy in a considerable proportion of treated subjects.

Last year, scientific community efforts in oncology were greatly conveyed to unhinge the reasons of this imbalance, trying to recognize the ideal candidate for immunotherapy; the final aim is to sharpen and potentiate ICIs tools [92]. ICIs activity is influenced by several clinical features, tumor histology, and also tumor genes’ expression [93].

The first studied marker has been PD-L1, a protein variously expressed by tumor cells, that for some cancer types correlate well with benefits from specific anti-PD-L1 agents [94]; in fact, some of these drugs have received approval on the basis of tumor PD-L1 expression [95]. For this reason, PD-L1 is to date the most used determinant in patients treated with ICIs worldwide, though it remains an imperfect predictive biomarker.

Evaluating US Food and Drug Administration (FDA) approvals of 45 ICIs agents across 15 tumor types, PD-L1 was found to be predictive in less than a third of them [94]. This phenomenon can be explained by multiple factors. First, PD-L1 expression in individual patients could be heterogeneous possibly being different in different tumor sites but also in a single tumor site, and influenced by the sample itself [96]. Secondarily, ICIs action can be affected by the huge amount of other tumor immune-microenvironment (TiME) properties [97]. B7-H3, a member of the B7 superfamily, is expressed on tumor cells and potentially circumvents CD8+-T-cell-mediated immune surveillance. Anti-B7-H3 immunotherapy combination with anti-PD-1/PD-L1 antibody therapy is a promising approach for B7-H3-expressing cancers [98]. Tumor-infiltrating T-cells (TILs) co-receptors (PD-1, CTLA-4, Lymphocyte-activation gene 3 (LAG-3), etc.) have great importance in the context of TiME, by interacting with tumor cells, becoming new target for a series of new ICIs, some of which already approved. Finally, factors related to the host immune system such as blood biomarkers (peripheral blood mononuclear cells (PBMCs), etc.) could affect ICIs activity [99].

Thus, it would be of extreme importance to consider TiME globally in patients eligible for ICIs. Analyses of TiME on tissue could be affected both by tumor tissue sample itself in terms of quantity and/or quality and by tumor heterogeneity [100]. NM imaging can bypass these two tissue intrinsic limits. Through specific radiotracer-labeled antibody, NM proved to characterize a series of TiME properties, and with good concordance with tissue data. Indeed, surface markers of PBMCss, TILs and tumor cells could be selectively targeted, leading to a successful definition of TiME composition with preclinical animal models, but also in human adults with cancer (Table 2). This constitutes a unique opportunity to overcome the intrinsic limits of TiME analyses on tissue, that results to be moreover both cost and time consuming.

Recently, imaging of TiME by PET has gained an excellent progress by the introduction of different tracers targeting fibroblast activation protein (FAP), among which ^68^Gallium-labeled chelator-linked FAP inhibitors (FAPIs) and ^18^F-labeled glycosylated FAPIs [119]. FAP is highly expressed by cancer-associated fibroblasts (CAFs), that are generally associated with poor prognosis and represent for some cancer types one of the main reason of resistance to immunotherapy [120]. 

NM imaging can provide a static baseline representation of the identified marker, so having a predictive value, possibly bringing our therapeutic choices to be as careful as possible. Otherwise, these techniques can be used in a more dynamic way, by repeating NM imaging in different disease time-points in patients treated with ICIs. The aim will be to monitor treatment response, by using labeled-radiotracers conjugated with immunotherapeutic agents, with first brilliant results in this sense.

### 3.1. PBMCs-Targeted Imaging

The generation of anti-CD8 ^89^Zr-desferrioxamine-labeled cys-diabody permitted a noninvasive immuno-PET tracking of endogenous CD8+ T cells in melanoma immunotherapy models, including anti-PD-L1 agents, antigen-specific adoptive T-cell transfer and agonistic antibody therapy.

The great power of such imaging consists in its ability to acquire dynamic information regarding therapy-induced alterations of both PBMCs and TILs component [101].

### 3.2. TILs-Targeted Imaging

Similarly, with an immuno-PET with Zirconium-89(^89^Zr)-labeled PEGylated single-domain antibody fragments specific for CD8, Rashidian et al. demonstrated the ability to longitudinally detect TILs in melanoma models and to distinguish responding tumors from those that do not respond to anti-CTLA4 therapy [102].

^123^I-interleukin-2 (IL-2) single-photon emission computed tomography (SPECT) well depicted TILs in cancer patients, also giving a preliminary response to cytokine treatment [103,104], while TILs targeting through ^99m^Tc-IL-2 in melanoma patients treated with I-O permits to acquire information about pseudo-progression and prognosis [105,106]. Prognosis impact has also found to be appraisable by quantifying TILs in animal models of bladder cancer trough anti-CD3 monoclonal antibody (mAb) modified with desferrioxamine (DFO) and radiolabeled with ^89^Zr [107]. Recognition of CTLA-4 TILs receptor has been realized by a specific ^64^Cu-DOTA-anti-CTLA-4 mAb [108,109], while anti-PD-1 agents labeled with ^64^Cu or ^89^Zr can detect very precisely PD-1-expressing TILs [110,111,112,113]. Moreover, the fully human anti-LAG3 antibody REGN3767, radiolabeled with ^89^Zr using the bifunctional chelator p-SCN-Bn-Deferoxamine (DFO) led to an immune-PET detection of LAG-3 expressing TILs [114]. NM individuation of PD-1, CTLA-4 and LAG-3 expressed by TILs could assist in patient stratification, disease diagnostics, and monitoring response to ICIs.

### 3.3. Tumor Cell-Targeted Imaging

A similar usefulness may derive from PET-imaging of some proteins expressed directly by tumor cells. High affinity to PD-L1 has been guaranteed by specifically synthetized ^68^Ga-labeled single-domain antibody tracer called ^68^Ga-NOTA-Nb109, and ^89^Zr-DFO-PD-L1 monoclonal antibody [115,116]. Moreover, saturation of B7-H3 tumor cell protein by anti-B7-H3 humanized monoclonal antibody DS-5573a labeled with ^89^Zr was demonstrated in tumors responding to DS-5573a therapy [117]. ^89^Zr-DFO-daratumumab provides successful whole-body PET visualization of myeloma, and predicts the effectiveness of daratumumab therapy [118].

All these applications represent in a certain way an evolution of the most widely used FDG-PET, already proved as seen to acquire response to ICIs earlier than through the classical CT scan imaging [72,78,79].

### 3.4. Tumor Heterogeneity

The therapeutic approach cannot ignore inter-patients and intra-patient cancer heterogeneity. The latter depends on a complex phenotypic evolution that involve (epi)genetic transformations, surface markers selections and cell-cycle modifications in response to environmental stimuli [121]. What is more, not only tumor cells but also TiME features might become different [122]. This is quite challenging because prognosis and response to treatments could be thus tightly influenced.

NM may be helpful to visualize tumor heterogeneity, with interesting applications in the field of I-O. Quantitative analyses of canonical ^18^F-FDG PET imaging can point out the presence of aggressive tumor foci, which may divert patients’ prognosis [123]; CAFs have found to contribute to the intratumor heterogeneity of ^18^F-FDG PET and to enhance tumor uptake [123].

The recognition of TiME and cancer cells markers heterogeneity, through specific tracers, could guide in the future daily therapeutic choices. PD-L1 is the forerunner marker in this sense, with different contributes about its possible discordant expression in different tumor lesions [124,125,126], but the same concept could be applied to other markers.

## 4. NM and Immune-Adverse Events (irAEs) Analysis

Immune-related adverse events (irAEs) are unique adverse events caused by activation of an immune response against healthy tissues due to immunotherapy. IrAEs occur in about two thirds of patients treated with IO, of which 14% is grade 3 or superior [127]. Timely detection and correct management of irAEs according to their grading, with suspension of IO or eventually immunosuppressive medication, are crucial [128,129]. If not treated, irAEs could require hospitalization and may lead to life-threatening complications. Therefore, predictive markers for irAEs are an unmet clinical need [130].

Diagnosis of irAEs is made through clinical, biohumoral, and radiological investigations. Early radiological detection of irAEs could improve their correct management. In a recent study, medical imaging detected 74% of irAE in patients treated with anti-PD1. Among typical irAEs, interstitial lung disease is mainly diagnosed with CT scans, as well as enterocolitis and pancreatitis, whereas thyroiditis with ultrasound and hypophysitis with magnetic resonance imaging [131].

Beyond tumor lesions, FDG uptake can be induced by the inflammatory reaction related to irAEs. Thus, sarcoid-like reaction [132,133] and enterocolitis [134] are easily detected by PET/CT scans. PET/CT could also be useful in detecting thyroiditis, hypophysitis, pancreatitis, gastroduodenitis, and cholangitis [131,135,136,137]. In another study with 35 patients with malignant melanoma receiving nivolumab and ipilimumab combination, 39% of irAEs were diagnosed subsequent to FDG-PET/TC and 11% remained clinically occult [138].

Furthermore, irAEs could predict response to immunotherapy [139]. A retrospective study enrolled 40 patients with malignant melanoma, malignant lymphoma or renal cell carcinoma; all patients with an increase of SUVmax in the thyroid of more than 1.5 on the first restaging scan compared to baseline scan had a complete response to I-O in 1 year [140]. Overt thyroid irAEs could be predicted by baseline thyroid uptake of FDG in another study that enrolled 200 patients treated with nivolumab. In the same study, thyroid irAEs were related to good prognosis in patients diagnosed with lung cancer [141]. As regards to enterocolitis, increased FDG uptake correlated with clinical symptoms but did not predict clinical outcome to ipilimumab in 100 advanced melanoma patients [142]. In a study with 41 patients with malignant melanoma treated with ipilimumab, 4 out of 31 patients having disease control were diagnosed sarcoid-like reaction in the follow-up PET/TC, while none of the patients with progressive disease [143].

## 5. Conclusions and Future Applications

NM imaging has historically contributed in a decisive way to expand the knowledge about cancer management and cancer response to treatment, providing several functional data. In the context of modern I-O, this could become quite challenging, because each day millions of cancer patients worldwide are treated with I-O therapies, and lots of crucial aspects remain to be clarified. Last decade scientific reports added progressively important pieces to the definition of the complex immune-cancer cycle, finding out favorable clinical and pathological patterns of response to I-O drugs, and even creating complex predictive algorithms.

Main open questions include the optimal candidate for I-O therapy, the determination of response and appropriate duration of treatments, the mechanisms that underlie primary and secondary resistance to I-O. The goals are I-O empowerment in terms of efficacy and healthcare resources optimization.

As seen, PET-CT scans has shown to predict responders and non-responders to I-O earlier than the conventional CT scan, through the acquisition of both anatomical and metabolic data. Moreover, several PET-CT scan parameters, acquired at baseline or during treatment, have been found to be predictive of response to I-O. NM applications could well depict several predictive TiME features by using different radiotracers and/or drug-radiotracer conjugates, and also precociously recognize drug-induced irAEs usually correlated with I-O benefit. This would permit ideally to change in the near future the approach of cancer care, by selecting the right I-O drug for the right patient, revealing the immunologic reasons of innate and acquired resistance to I-O, and laying the foundations to get around them.

In the last few years, I-O imaging field has been enriched by radiomics, a set of methods used to extract a large amount of quantitative data from biomedical images—called features—aiming to personalize medicine even more. The typical radiomics workflow is a multi-step process, that starts with the acquisition of measurable features and, through different phases of data processing, concludes with data analysis [144]. Radiomics can acquire and deal with traditional radiology imaging (CT, magnetic resonance), but also with PET imaging, building-up mathematical algorithms in order to study tumor characteristics and clinical outcomes. PD-L1 expression of patients affected by head and neck carcinoma (n = 53) has found to be correlated to textural features derived from FDG-PET organization of tumor pixels [145]. PET/CT-based radiomic signature could have both a prognostic value for NSCLC patients that underwent to surgery [146] and a predictive role to identify those patients most likely to benefit from immunotherapy. A multiparametric radiomics signature, built up combining different radiomics features, have shown in fact to well predict durable clinical benefit from immunotherapy in advanced NSCLC patients (n = 99) [147], with other experiences confirming this potentiality [148]. Moreover, radiomics could improve tumor heterogeneity characterization and prognosis prediction, in order to stratify patients to receive therapeutic regimens [123]. Despite its great premises, some concerns exist about applicability of NM radiomics in daily clinical practice [149], and further studies investigating its potential in I-O field are urgently warranted.

## Figures and Tables

**Figure 1 cancers-12-01303-f001:**
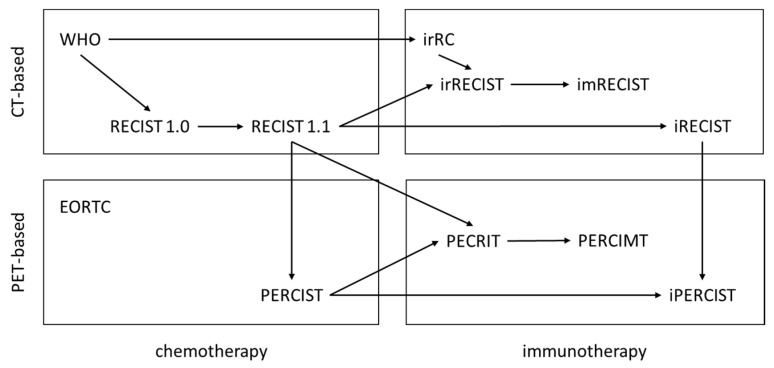
Overview of computed tomography (CT)-based (upper panel) and positron emission tomography (PET)-based (lower panel) response criteria for solid tumors. On the left side are grouped response criteria designed for chemotherapy, on the right side those specifically designed for immunotherapy. RECIST: Response Evaluation Criteria in Solid Tumors, irRC: Immune Related Response Criteria, irRECIST: Immune-related Response Evaluation Criteria in Solid Tumors, imRECIST: Immune-modified response evaluation criteria in solid tumors, iRECIST: Immune response evaluation criteria in solid tumors. EORTC: European Organization for Research and Treatment of Cancer criteria, PERCIST: PET Response Criteria in Solid Tumors, PERCIMT: PET Response Evaluation Criteria for Immunotherapy, PECRIT: PET/CT Criteria for Early Prediction of Response to Immune Checkpoint Inhibitor Therapy, iPERCIST: immune PET Response Criteria in Solid Tumors.

**Table 1 cancers-12-01303-t001:** Immune checkpoint inhibitor for solid tumors (European Medicines Agency and Food and Drugs Administration approval).

Immune Checkpoint Inhibitor	Solid Tumor	Reference
Adjuvant Setting		
Ipilimumab	Malignant melanoma	[6]
Nivolumab	Malignant melanoma	[7]
Pembrolizumab	Malignant melanoma	[8]
Advanced Disease Setting		
Ipilimumab	Malignant melanoma	[9]
Nivolumab	Malignant melanoma	[10]
	Non-small cell lung cancer	[11,12]
	Renal cell carcinoma	[13]
	Hodgkin’s lymphoma	[14]
	Head and neck cancer	[15]
	Urothelial carcinoma	[16]
	Mismatch-repair deficient/Microsatellite instability-high colorectal carcinoma ^1^	[17]
	Hepatocellular carcinoma ^1^	[18]
Nivolumab plus Ipilimumab	Malignant melanoma	[19]
	Renal cell carcinoma	[20]
	Mismatch-repair deficient/Microsatellite instability-high colorectal carcinoma ^1^	[21]
Pembrolizumab	Malignant melanoma	[22]
	Non-small cell lung cancer (with or without chemotherapy)	[23,24,25]
	Renal cell carcinoma (with axitinib)	[26]
	Hodgkin’s lymphoma	[27]
	Head and neck cancer ^1^	[28]
	Urothelial carcinoma	[29,30]
	Hepatocellular carcinoma ^1^	[31]
	Gastric cancer ^1^	[32]
	Esophageal cancer ^1^	[33]
	Cervical cancer ^1^	[34]
	Merkel cell carcinoma ^1^	[35]
	Small cell lung cancer ^1^	[36]
Atezolizumab	Urothelial carcinoma	[37,38]
	Non-small cell lung cancer (with or without chemotherapy and bevacizumab)	[39,40,41]
	Small cell lung cancer (with carboplatin and etoposide) ^1^	[42]
	Triple-negative breast cancer (with nab-paclitaxel) ^1^	[43]
Avelumab	Merkel cell carcinoma	[44]
	Urothelial carcinoma ^1^	[45]
	Renal cell carcinoma (with axitinib) ^1^	[46]
Durvalumab	Non-small cell lung cancer	[47]
	Urothelial carcinoma ^1^	[48]

^1^ Food and Drugs Administration approval only.

**Table 2 cancers-12-01303-t002:** Tumor immune-microenvironment analysis with nuclear medicine imaging.

TiME Type	Target	Radionuclide	Cancer Type	Setting	Comment	Reference
PBMCs	CD8	^89^Zr-anti-CD8	Melanoma cell lines	Preclinical syngeneic tumor	Detecting change in systemic CD8+ T-cells	[101,102]
TILs	IL-2 receptor	^123^I-IL-2	SCCHN	Human	–	[103]
			RCC	Human	Identify patients that less likely will benefit from cytokine treatments	[104]
		^99m^Tc-IL-2	Melanoma	Human	Prognostic information	[105]
			Melanoma	Human	–	[106]
	CD3	^89^Zr	Bladder cancer lines	Bearing mice	DFO-anti-CD3 had diminished CD4+ T-cell counts and polarization of the CD8+ T-cell pool towards a memory phenotype	[107]
	CTLA-4	^64^Cu-DOTA-anti-CTLA-4	Colon cancer cell lines	Bearing mice	–	[108]
		^64^Cu-DOTA-ipilimumab	Lung cancer cell lines	In-vitro and in-vivo (bearing mice)	–	[109]
	PD-1	^89^Zr-Df-nivolumab	Lung cancer cell lines	In-vitro and in-vivo (bearing mice)	–	[110]
		^64^Cu-labeled PET	Melanoma cell lines	Bearing mice	Images of FoxP3(+) CD4(+) Tregs	[111]
		^64^Cu-pembrolizumab	Melanoma cell lines	Bearing mice	–	[112]
		^89^Zr-pembrolizumab	Melanoma cell lines	Bearing mice	Clinically translatable to monitor cancer response to ICIs	[113]
	LAG-3	^89^Zr-REGN3767	Lymphoma cell lines	Bearing mice	–	[114]
Tumor cells	PD-L1	^89^Zr-DFO-PD-L1 mAb	Breast, gastric, lung cancer cell lines	In-vitro and in-vivo (bearing mice)	Uptake increased with escalating dose of avelumab	[115]
		^68^Ga-DOTA-Nb109	Melanoma cell lines	Bearing mice	–	[116]
	B7-H3	^89^Zr-DS-5573a	Breast, colon cancer cell lines	Bearing mice	Identify tumor responding to therapy, insight into T cell biology	[117]
	CD-38	^89^Zr-daratumumab	Myeloma cell lines	Bearing mice	Predict effectiveness of daratumumab	[118]

The above table depicts the advancements of NM in the determination of the different TiME features. For each TiME target, it is specified the radionuclide or the antibody-drug conjugates used, the clinical or preclinical setting, the type of cell line. Finally, any other main information in terms of recognition of target, determination of treatment response is reported as comment. CD: cluster of differentiation; CTLA-4: cytotoxic T-lymphocyte antigen 4; Df: p-SCN-deferoxamine; DFO: desferrioxamine B; DOTA: tetra-azacyclododecanetetra-acetic acid; FoxP3: forkhead box P3; ICI: immune checkpoint inhibitors; IL-2: interleukin-2; LAG-3: lymphocyte-activation gene 3; PBMCs: peripheral blood mononuclear cells; PD-1: programmed cell death protein 1; PD-L1: programmed cell death protein-ligand 1; RCC: renal cell carcinoma; SCCHN: squamous cell carcinoma of the head and neck; TILs: tumor-infiltrating lymphocytes; Treg: Regulatory T cell; ^64^Cu: copper-64; ^68^Ga: gallium-68; ^123^I: iodium-123; ^99m^Tc: technetium-99m; ^89^Zr: zirconium-89.

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
