# Peer review of "Novel Nuclear Medicine Imaging Applications in Immuno-Oncology"

_cancers, 2020, doi:10.3390/cancers12051303_

Round 1
Reviewer 1 Report
The manuscript by Frega and coworkers entitled „Novel nuclear medicine imaging applications in Immuno-Oncology” is a review article concerned with an overview of the implemented PET imaging studies in the field of immune-oncology.
The paper is well written and for the field of nuclear medicine very valuable, to gain an overview about the use of FDG-PET imaging in response to immunotherapy studies. The review of the FDG-related studies is adequately presented and main results are discussed adequately.
However, there are two major points of criticism (which are in fact easy to correct and thereby add value to the manuscript) which should be addressed by the authors when they revise their manuscript :
- Page 7, line 218-228. Imaging of the microenvironment of tumors by PET has gained an excellent progress by the introduction of 68Ga-FAPI-04 of 18F-FGlc-FAPI , recently published by Lindner JNM 2018 or Toms JNM 2020. Such tracers could be highly useful in future imaging studies of fibroblast activation protein expressed on cancer-associated fibroblasts (CAFs), since their presence are frequently the main reason for unefficient immunotheraphy, for example in pancreatic cancer!
This reviewer suggests to rewrite the sentence on page 7, line 227 (“PET imaging agents can be considered …..” , in order to be more precise for the reader , the authors could refer the reader to the above-mentioned tracers. This would provide the reader with information about the future needs in nuclear medicine imaging applications. - Page 7, line 233/234: “The aim will be to monitor treatment response …… with first brilliant result … [101-110].”The authors should add a separate short paragraph to deal with these very interesting studies in the review article in much more detail.
The manuscript of the authors almost exclusively discussed clinical results from FDG-PET studies. As noticed by the authors, there a brilliant studies (and the author cite 10 of these papers!) that are really of very high quality, and that need to be discussed in much more detail in the revised version of this review article !
Reviewer 2 Report
This is an interesting review article by Frega et al. on the role of PET imaging in immune-oncology with focus on newer, mostly ICI-based approaches. The authors present the main data on this rapidly growing, revolutionary field in a concise way. The manuscript is well-written. Although, similar review articles have been published on the topic, this work represents a very satisfying attempt to present the existing knowledge and future perspectives of nuclear medicine in immunotherapy.
I believe that the manuscript could be accepted after the following points are addressed:
- A more detailed description of the limitations/drawbacks of PET imaging with the tracer FDG should also be made.
- Figure 1 is rather poor graphically, and not absolutely correct. For example the PECRIT criteria are related more to EORTC than PERCIST.
- The link to the Supplementary material is not functioning.
- Table 2: the authors could also consider mentioning the agent 89Zr-Daratumumab on multiple myeloma with satisfying results of a Phase I first-in-human study. (Ulaner, G.; Sobol, N.; O'Donoghue, J.; Burnazi, E.; Staton, K.; Weber, W.; Lyashchenko, S.; Lewis, J.; Landgren, C.O. Preclinical development and First-in-human imaging of 89Zr-Daratumumab for CD38 targeted imaging of myeloma. J. Nucl. Med, 2019, vol. 60 no. supplement 1 203. SNMMMI Annual Meeting).
- The names in Reference 78 seem wrong. Please correct.
Reviewer 3 Report
This review by Frega et al. gives a nice overview of the state of the art on the evaluation criteria for immunotherapy and on the increasing importance of nuclear medicine in this field.
MAJOR COMMENTS
1) In the abstract it is stated that the manuscript is a systematic review. However, objectives and eligibility criteria for studies are missing. The same applies to searching methodology and risk of bias assessment. To me this review does not seem to be systematic.
2) The 2.0 immuno-oncology section is very compressed. The radiopharmaceuticals reported in table 2 are described with one sentence: “The aim will be to monitor treatment response, by using labeled-radiotracers conjugated with immunotherapeutic agents, with first brilliant results in this sense”.
I suggest to expand this section and discuss in details each mentioned radiopharmaceuticals, maybe with pros and cons over [18F]FDG.
The use of targeted radiopharmaceuticals has great potential and is a field in expansion.
3) I foresee that one of the major challenges to evaluate immunotherapy will be tumor inter/intra patient heterogeneity. Indeed, it has been published that different lesions, in the same patient, can express different markers or be in a different differentiation state.
Authors discuss this topic in details.
MINOR COMMENTS
Page 2 – line 30 (and following pages) 18-FDG PET/CT should be replaced by either [18F]FDG PET/CT or 2-[18F]FDG (which is the correct one according to EANM guidelines).
Page 3 – line 77. The sentence: “Basically, overall tumor burden is quantified by summing the size of tumor lesions in a baseline 77 computed tomography (CT) scan before the start of a new therapy.” is copy pasted from: “Evaluating tumor response with FDG PET: updates on PERCIST, comparison with EORTC criteria and clues to future developments” By Katja Pinker, Christopher Riedl, and Wolfgang A Weber. Please change it.
Page 3 – line 88. Correct nomenclature is 2-deoxy-2-[18F]fluoro-D-glucose or 2-[18F]fluoro-2-deoxy-D-glucose
Table 2. TILs studied with 99mTc-IL2 in melanoma. Two more references should be added:
Markovic SN, Galli F, Suman VJ, et al. Non-invasive visualization of tumor infiltrating lymphocytes in patients with metastatic melanoma undergoing immune checkpoint inhibitor therapy: a pilot study. Oncotarget. 2018;9(54):30268–30278. Published 2018 Jul 13. doi:10.18632/oncotarget.25666
Reviewer 4 Report
This review article by Frega and colleagues tackles an important subject of the role of molecular imaging in immunotherapy. The article provides a summary of immunotherapy, briefly reviews the development of anatomical imaging response criteria, the adaptation of immunotherapy-modified response assessment, and then discusses the role of FDG PET and novel agents targeting immune microenvironment. However, the article does not clearly define the shortfalls of anatomical imaging which would be a prelude to the section on response assessment and TiME imaging. The sections on FDG PET is also brief and not conclusive. Besides, several manuscript edits, clarifications, and grammar mistakes need to be addressed. These are as follows:
Abbreviation of tumor immune microenvironment needs to be consistent throughout the test (TiME or TIME)
Line 66, “It is then crucial identifying and developing” to identify and develop
Line 121, “… great potential to early recognize response”. … for early recognition of response …
Line 121-122, “ The 18F-FDG PET/CT has the great potential to early recognize response to immunotherapy, and the decrease of tumor parameters at early time-point PET scans was seen in patients treated with ICIs who will have later clinical benefit” provide the reference
Line 124, “basal” change to baseline. This needs to be corrected throughout the manuscript
The paragraph starting 125, this paragraph is not focused. Mainly early metabolic response is discussed but, in the middle,, there is a study indicating role of FDG PET at 1 year. Please revise.
Paragraph starting line 132, “Likewise, semi quantitative parameters were found to be predictive of outcome in patients treated with immunotherapy. Apart from SUV parameter, by the time other indexes became relevant, such as metabolic tumor volume (MTV) and tumor total lesion glycolysis (TLG), that indicate a measurement of the tumor volume with a high metabolism and the product of the mean SUV and the MTV, respectively. These three variables well define the entire metabolic tumor burden (MTB), that may represent a reliable value to decree treatment success, being known that I-O work better in a context of low tumor load [73].” This paragraph is hard to follow and needs revision
Line 140, “at CT scan” change to by CT scan
Line 164,: “sensibility” change to sensitivity
Line 169, “…iPERCIST was validated “criteria were not validated. Criteria were proposed.
Line 190, “ Cancer patients experience different range of benefit from the use of ICIs in terms of survival.” Please revise this sentence.
Line 192, “ … survival perspective for a quote of patients, … “ ? quote
Line 196, “ … in optic to …” not sure what this means
Line 203, ” For several cancers, PD-L1 expression do not match with survival prolongation, with PD-L1 negative ones responding or at the opposite high PD-L1 ones progressing to ICIs” Please revise this sentence
Line 206, “ … cancer patient PD-L1 expression” can change to “PD-L1 expression in individual patients …”
Line 209, “ … properties, inside the so complex tumor cell immunity cycle “ revise this statement
Line 217, “… could be a discriminant for the profitable action of ICIs” revise please
Line 218, “ In this scenario, NM application could deserve an interesting place in the 2.0 immuno-oncology “ I am unsure what this sentence conveys. This aim of the entire paragraph is not clear.
Line 233, “… The aim will be to monitor treatment response, by using labeled-radiotracers conjugated with immunotherapeutic agents, with first brilliant results in this sense” elaborate further what brilliant results refer to
Line 235, “ These represents ..” represent
Line 258, “ bioumoral and ..” is this a typographical error?
Line 273, “ By basal thyroid uptake of FDG-PET, overt thyroid irAEs could be predicted therapy in another study enrolling 200 patients treated with nivolumab.” Revise this sentence.
Line 275, “As regards enterocolitis …” as regards to…
Line 290, “to I-O. To widen the discovery in this sense would be useful, thus driving towards an I-O empowerment in terms of efficacy, and towards healthcare resources optimization. PET-CT scans performed at early time points well predict benefit from I-O, so anticipating what the most classically used CT scan could allow” Revise and simplify these sentences.
Lind 293, “ Furthermore, thanks to the acquisition of both anatomical and metabolic data, NM could be helpful to untangle some gray areas like pseudoprogression.” There is not enough data presented to justify this statement.
Line 296, “Moreover, benefit from I-O could correlate with some kind of drug-induced irAE, that NM imaging is able to well describe. Finally, through the application of different radiotracer and/or drug-radiotracer conjugates, NM could depict TiME composition all-round.” Revise these sentences.
Line 299, “ next future” ? near future
Line 300, “adaptive and acquired” these two have a similar meaning
Table 2. The comment column is not well aligned with the respective rows and difficult to follow.
Reference 78 appears to be incorrect
Round 2
Reviewer 3 Report
Manuscript improved.
All criticisms raised by reviewers were taken into consideration.
Reviewer 4 Report
In this revision, the authors have responded to the reviewers' comments appropriately.